# Transcriptome Analysis Reveals HgCl_2_ Induces Apoptotic Cell Death in Human Lung Carcinoma H1299 Cells through Caspase-3-Independent Pathway

**DOI:** 10.3390/ijms22042006

**Published:** 2021-02-18

**Authors:** Mi Jin Kim, Jinhong Park, Jinho Kim, Ji-Young Kim, Mi-Jin An, Geun-Seup Shin, Hyun-Min Lee, Chul-Hong Kim, Jung-Woong Kim

**Affiliations:** Department of Life Science, Chung-Ang University, Seoul 06974, Korea; mjkim831025@gmail.com (M.J.K.); shoopjh@cau.ac.kr (J.P.); jinhokim07@cau.ac.kr (J.K.); jykim@cau.ac.kr (J.-Y.K.); dksalwls333@gmail.com (M.-J.A.); rjstjq89@naver.com (G.-S.S.); lhmscb@naver.com (H.-M.L.); 01031e70067@gmail.com (C.-H.K.)

**Keywords:** mercury chloride, cell cycle arrest, apoptosis, caspase-3, RNA-seq, human non-small cell lung carcinoma cells, H1299

## Abstract

Mercury is one of the detrimental toxicants that can be found in the environment and exists naturally in different forms; inorganic and organic. Human exposure to inorganic mercury, such as mercury chloride, occurs through air pollution, absorption of food or water, and personal care products. This study aimed to investigate the effect of HgCl_2_ on cell viability, cell cycle, apoptotic pathway, and alters of the transcriptome profiles in human non-small cell lung cancer cells, H1299. Our data show that HgCl_2_ treatment causes inhibition of cell growth via cell cycle arrest at G_0_/G_1_- and S-phase. In addition, HgCl_2_ induces apoptotic cell death through the caspase-3-independent pathway. Comprehensive transcriptome analysis using RNA-seq indicated that cellular nitrogen compound metabolic process, cellular metabolism, and translation for biological processes-related gene sets were significantly up- and downregulated by HgCl_2_ treatment. Interestingly, comparative gene expression patterns by RNA-seq indicated that mitochondrial ribosomal proteins were markedly altered by low-dose of HgCl_2_ treatment. Altogether, these data show that HgCl_2_ induces apoptotic cell death through the dysfunction of mitochondria.

## 1. Introduction

Since the development of the industry, people worldwide have faced serious environmental pollution. Heavy metals, including mercury and lead, are one of the toxic elements found in environmental pollutants [1]. Among them, mercury is the third most harmful toxicant on the planet, according to the US Government Agency of the most toxic substances [2], and WHO (World Health Organization) regards mercury to be one of the top 10 chemicals of major public human health [3]. 

Mercury naturally exists at low concentrations in air, water, and soil. However, mercury accumulates abnormally in the atmosphere and soil as a result of human population growth and its activities. The anthropogenic release of mercury poses a high risk to human health [4]. Human exposure to toxicants is mainly through air pollution, and oral exposure via contaminated water and food is also an important route [2,4]. In addition, inorganic mercury has been used in personal care products such as dermatological lotions, drugs, and germicide substances, resulting in continuous exposure of the human being to their toxic effects [5]. Mercury exists generally in inorganic (mercury chloride, HgCl_2_) and organic mercury (methyl mercury, MeHg). Chemical forms of mercury directly affect their toxic properties, toxicokinetics, biological behavior, and clinical manifestations [2,5]. Currently, most of the studies focused on the toxicity of methylmercury due to its relative stability, effective absorption, and long half-life for digestion. However, previous studies have reported that inorganic mercury exists relatively more in environmental conditions such as sediments and ecosystems compared to organic mercury [6,7].

Exposure to inorganic mercury may cause serious damage to numerous organs in humans and animals. Because inorganic mercury has been reported that its toxic effect is much more perpetual than those of methylmercury [8]. Mercury chloride, which is inorganic mercury, can cause the occurrence of disease in humans due to critical organ damage including liver [9], kidney [10], central nervous system [11], cardiovascular system [12], and reproductive system [13]. In our previous study, we also demonstrated the adverse effect of mercury chloride on cell life in the human placenta [14]. Like this, studies involving the toxic effects of mercury chloride in humans and animals are well known in the literature. However, little is presently known of the changes of gene expression in lung cells with exposure to mercury chloride. Thus, we investigated to identify the effects of mercury chloride with a low dose on transcriptome profiling using RNA-sequencing in human non-small-cell lung cancer cell line, H1299. We also demonstrated the effect of mercury chloride on cell viability, cell cycle, as well as apoptosis.

## 2. Results

### 2.1. The Effect of HgCl_2_ on Cell Viability and Growth in Human Non–Small-Cell Lung Cancer Cell Line, H1299

To investigate the cellular physiological effects of HgCl_2_ treatment, H1299 cells were treated with different concentrations of HgCl_2_ for 24 and 48 h. First, we tested the cell viability using a MTS (Tetrazolium inner salt) assay. As shown in Figure 1A, exposure to HgCl_2_ significantly decreased the viability of H1299 at high concentrations (50–100 μM). Compared to the control group, HgCl_2_ reduced the cell viability, with an IC_50_ value of 36.2 μM at 48 h incubation. The cell viability test was further measured using the live or dead assay with H1299 cells. For this, we treated various concentrations of HgCl_2_ (0–20 μM) in H1299 cells. However, no results did not reveal in this dose range (Appendix A). Based on these results, we selected an effective concentration range of HgCl_2_ treatment (40, 60 μM) for the live or dead assay and the subsequent experiments. Concordance with Figure 1A, HgCl_2_ treatment strongly increased the number of dead cells at 60 μM for both time points, but no effect was detected with 40 μM exposure (Figure 1B). We also observed the morphological changes by HgCl_2_ treatment using phase-contrast microscopy. HgCl_2_ treatment showed morphological features corresponding to apoptotic cell death such as cellular shrinkage, forming apoptotic bodies in a dose- and time-dependent manner (Figure 1C), suggesting HgCl_2_ has cytotoxic effects in human non-small-cell lung cancer cell line, H1299. To further investigate whether HgCl_2_ affected cell proliferation, cells were immunostained with specific antibodies against Ki-67, which is a cell proliferation marker, and analyzed using flow cytometry. HgCl_2_ treatment significantly elevated cell proliferation in a dose- and time-dependent manner, contrary to our expectation. Especially, the proportion of Ki-67 positive cells were raised at 60 μM treatment of HgCl_2_ compared to the control (Figure 1D).

### 2.2. HgCl_2_ Treatment Caused Cell Cycle Arrest via Dysregulation of Cell Cycle-Related Protein, Cyclin B1 and Cyclin D1 in H1299 Cells

To determine whether the cell cycle progression of H1299 by HgCl_2_ treatment, cells were treated with different concentrations of HgCl_2_ (40, 60 μM) for indicated time points. Exposure to HgCl_2_ with 60 μM revealed a significant cell cycle arrest at G_0_/G_1_- and S-phase at 24 h (1.7-fold at both phases), but 48 h incubation showed decreasing the number of PI (propidium iodide)-positive cells in all progress (Figure 2A). However, cell cycle progression did not change by HgCl_2_ treated with 40 μM. We next investigated molecular mechanisms underlying the cell cycle arrest by HgCl_2_. It is well known the various biomarkers such as cyclins and CDKs (cyclin depedent kinases) are involved in cell cycle progression [15]. Among them, we identified the expression patterns of cyclin B1 and cyclin D1 by HgCl_2_ in the present study.

As shown in Figure 2B, the expression level of Cyclin B1, which is a marker for G_0_/G_1_, was significantly increased at the high dose (60 μM) of HgCl_2_ treatment, while HgCl_2_ treatment with 40 μM did not affect (Figure 2B). In addition, Cyclin D1 expression level, which is a marker for G_2_/M-phase, was increased at 40 μM treatment of HgCl_2_ for 24 and 48 h, but 60 μM treated with HgCl_2_ strongly reduced (Figure 2C). These results revealed that HgCl_2_ treatment arrests cell cycle progression through the alteration of cyclin B1 and D1 expression.

### 2.3. HgCl_2_ Induces Apoptotic Cell Death via The Caspase-3-Independent Pathway in H1299 Cells

To investigate whether HgCl_2_ causes apoptosis in H1299 cells, we assessed the induction of cell death by HgCl_2_ using Annexin-V and PI staining. Numerous studies have reported that mercury causes apoptotic cell death [11,14,16,17]. As shown in Figure 3A, the population of early- and late-apoptotic cells was significantly increased after HgCl_2_ treatment. Approximately, 95% of H1299 cells were double-positive for Annexin-V and PI showing late-apoptosis, with 60 μM of HgCl_2_ at 24 and 48 h (Figure 3A). Next, to further demonstrate the mechanisms underlying apoptosis induced by HgCl_2_, we tested the role of HgCl_2_ in the caspase-3 activation using flow cytometry. Caspase-3 activity was assessed by immunostaining with cleaved caspase-3 antibody, an active form of caspase-3. Interestingly, HgCl_2_ treatment did not show a dramatic increase in cleaved caspase-3 expression (Figure 3B and Appendix A), despite showing significance in quantitative values (Figure 3B). These results indicated the induction of apoptosis by HgCl_2_ was independent of the caspase-3 pathway.

### 2.4. Transcriptome Dynamics of HgCl_2_-Treated Human Non–Small-Cell Lung Cancer Cell Lines, H1299

To investigate the effect of HgCl_2_ exposure on transcriptional changes in H1299 cells, we selected the low concentration (5 μM) of HgCl_2_ that showed ineffective in cell death and physiology, and the cells subjected to directional RNA-seq analysis. For this, total RNA was isolated after HgCl_2_ treatment for 48 h, and libraries for mRNA-sequencing were constructed from the independent biological replicates. After filtering the RNA-seq result (FPKM (Fragments per kilobase of transcript per million reads) > 1, at least in one group), a comprehensive list of 19,014 transcripts was obtained (Appendix A). The two data sets showed concordant transcriptome dynamics in the HgCl_2_-exposed and control group. To identify congruency between each biological replicate, the principal component analysis (PCA) on transcripts using DEseq2 indicated that HgCl_2_ treatment estimated for the largest variance among all datasets generated using the RNA-seq platform. All the biological replicates from each experimental group were clustered (Figure 4A), identifying high reproducibility between the replicates. To test the expression pattern of mRNA transcript during exposure to HgCl_2_ in H1299 cells, heat maps were constructed to profile the differential expression pattern of entire transcripts at 48 h. Unsupervised hierarchical clustering analysis based on Pearson’s correlation of averaged and log2 of normalized FPKM values of the HgCl_2_-exposed and control groups revealed a critical shift in the HgCl_2_-treatment transcriptome in the form of up-and down-regulated transcripts (Figure 4B).

As shown in Figure 4C, the volcano plot was constructed by integrating both the P-value and fold change of each transcript (*p*-value ≤ 0.01 and absolute log2 [fold change] ≥1) to indicate the general scattering of the transcripts and to filter the differentially expressed transcripts for the HgCl_2_-treated group (Figure 4C). Application of DEseq2 with a conservative approach to the RNA-seq data obtained from the HgCl_2_-treated group identified 1624 differentially expressed (DE) transcripts (RNA-seq FPKM values having log2-fold change ≥ 1, adjusted *p*-value < 0.05, an average of FPKM in each group ≥ 20%). All 1624 DE transcripts (1191 up- and 433 downregulated transcripts) in the HgCl_2_-treated cells are listed in Appendix A. The result was verified by the heatmap expression based on the hierarchical clustering of expression ratios for the DE transcripts (Figure 4E).

### 2.5. Gene Ontology (GO) Indicated That HgCl_2_ Exposure Altered Gene Expression Sets Related to Cellular Metabolisms

To identify the transcriptomic pathway affected by HgCl_2_ exposure, the subset of DE transcripts that were significantly influenced by the HgCl_2_ treatment were subjected to GO annotation using Gorilla (http://cbl-gorilla.cs.technion.ac.il; finally accessed on 1 December 2020). To identify enriched biological associations among the up-or down-regulated DE transcripts, induced by exposure to HgCl_2_ (Figure 5). For each category, these results were identified to be statistically significant at *p*-value < 0.001. The up- and downregulated gene were independently subjected to GO analysis to discriminate them according to their functional roles. Enriched GO terms in ranked lists of genes by GOrilla were visualized using REVIGO bioinformatics resource (http://revigo.irb.hr; finally accessed on 1 Decemebr 2020). The biological processes which were significantly enriched were mostly involved in the cellular nitrogen compound metabolic process, cellular metabolism, and translation of gene expression categories were significantly clustered (Figure 5). Although we did not show in the main figure, most of the molecular functions have found in the RNA binding and structural constituent of ribosome (Appendix A).

## 3. Discussion

This study verified that HgCl_2_ presents a cytotoxic effect on human lung cells, subsequently leading to apoptotic cell death via the caspase-3 independent pathway. We further investigated the transcriptome profiling of H1299 cells exposed to HgCl_2_ with low concentration. In particular, HgCl_2_ has shown to involve the dysfunction of the metabolic process.

Mercury is a widespread toxicant, and causes concern as an environmental pollutant, and multiple forms of mercury exist in the environment, with different levels of cytotoxicity and implications for human health. Exposure to mercury can cause serious health risks affecting the nervous, cardiovascular, immune, and reproductive systems. In addition, mercury also causes detrimental effects on neuronal development and brain function [11,18]. For the last few decades, the EU banned several products to reduce the emissions from all relevant industrial activity. However, mercury is still used in various products, including numerous medications, dental components, fungicide agents, and personal care products [17]. Air pollution is the most common cause of human exposure to environmental pollutants such as mercury [19], however, most people are exposed to mercury through consuming contaminated food or water. According to the WHO, the prevalence of various respiratory diseases such as asthma, bronchitis, and lung cancer caused by air pollution is rapidly increasing [3].

The aim of the study was to evaluate the cytotoxicity of HgCl_2_ through in vitro experiments and to measure alters on gene expression induced by HgCl_2_. Although most of the studies focused on the adverse effect of organic mercury in various literature, we showed the harmful effect of HgCl_2_ on human lung cell lines, H1299 in present study. Here, we found that HgCl_2_ induced the suppression of cell growth and apoptosis in H1299 cells. Despite increasing the inhibition of cell death, HgCl_2_ increased the expression levels of Ki-67 in H1299. Interestingly, in the previous study, we obtained a similar result in the analysis of Ki-67 expression affected by environmental toxicant [20]. In that study, methylparaben which is one of the toxic environmental pollutes, increased the expression levels of Ki-67 in H1299 cells. In addition, several studies have shown that Ki-67 has different outcomes depending on the cell line [21]. For example, Ki-67 is required for the regulation of cell cycle progression in human and mouse cells [22,23]. In contrast, it has also been reported that inhibition of Ki-67 did not show changes in cell cycle distribution and cell proliferation in vitro and in vivo [21,24,25]. These studies suggested that the Ki-67 regulates the cell cycle progression depending on cell-type specificity.

Several studies have demonstrated the biological mechanisms proposed to explain the cytotoxic effect of HgCl_2_. It has been well known that oxidative stress induced by HgCl_2_ is considered to be one of the causes associated with cell death [26,27,28]. The inorganic mercury, such as HgCl_2,_ has a prominent affinity for thiol (-SH) groups of endogenous biomolecules, such as glutathione (GSH) and sulfhydryl proteins [16]. Consequently, inorganic mercury decreased the GSH level and provoked the induction of oxidative stress [29]. It is well known that oxidative stress is a critical mediator in diverse forms of apoptosis [30]. The mechanisms of the cell death signaling pathway are well established by mercury in several in vivo and in vitro experiments [11,26,29,31]. Although the generation of oxidative stress by mercury was not identified in this study, we determined that the molecular mechanism of apoptosis induced by HgCl_2_ in human lung cells was elucidated (Figure 3). Here, we found that mercury induced the suppression of cell viability and increased apoptotic cell death. However, despite causing increased apoptosis, the mercury did not change the activation of caspase-3 in H1299 cells (Figure 3). Like our results, several studies have shown that mercury induces apoptosis via a downstream caspase-independent pathway. For example, mercury induced the dysfunction of mitochondria in mouse cerebellar granule cells [32]. In addition, deregulated Gsk3 (Glycogen synthase kinase 3) activity, which is involved in the intrinsic apoptotic pathway, by mercury chloride revealed the liver injury in zebrafish [33]. Furthermore, cell death induced by mercury was also caused by the caspase-3 pathway. Teixeira et al., have suggested that exposure to mercury causes apoptosis by caspase-3 activation depending on the experimental conditions in the motor cortex [11]. 

Finally, to investigate the diverse biological responses to HgCl_2_, we analyzed comparative transcriptome profiling by RNA-seq performed in H1299 cells treated with low-dose (5 μM) of HgCl_2_ for 48 h. Our findings from gene expression profiling of human lung carcinoma cells exposed to HgCl_2_ suggest that HgCl_2_ interrupts the normal function of cellular metabolism. Interestingly, we found the mitochondrial ribosomal protein s18 (MRPS18), 19, 22, 25, 28, and mitochondrial ribosomal protein l13 (MRPSL13), 16, 32, 46 were expressed in the upregulated genes related to cellular metabolisms. In contrast, the identified genes, including mitochondrial ribosomal protein s14 (MRPS18), 15, and 21, were downregulated by exposure to HgCl_2_ in H1299 cells. Recent findings regarding the mitochondrial ribosomal protein reported that the dysregulated MRPs (Mitochondrial ribosomal proteins) affects mitochondrial metabolism disorder, cell dysfunction leading to apoptosis [34]. Furthermore, mercury ions specifically blocked synthesis of ribosomal RNA whereas activity of RNA polymerase II remained unchanged [35]. In the muscular tissues of two species of fish (Prochilodus lineatus and Mylossoma duriventre) that exposured to the mercury ions showed enhanced expression levels of 40S ribosomal protein S27a [36]. This suggests that the abnormal regulation of mitochondrial ribosomal and cellular ribosomal protein-related genes by HgCl_2_ may accelerate apoptosis of H1299 cells. In previous studies, we investigated whether the environmental chemical, mercury chloride, causes apoptotic cell death in BeWo cell without transcriptome profiling [14]. In addition, we also demonstrated the comparative analyses of comprehensive gene expression of H1299 cells treated with methylparaben, which is one of the environmental toxicants [18]. Methylparaben was shown to change gene expression through alternative splicing events. Here, we focused on comprehensive transcriptome profiling of HgCl_2_ in H1299 cells, as this is a critical phenomenon affected by low-dose of HgCl_2_ exposure. The altered gene expression pattern will be a potential strategy for the diagnosis and treatment of diseases caused by environmental toxic substances.

In conclusion, we identified that HgCl_2_ induces the inhibition of cell viability and increasing apoptotic cell death. Cell death of H1299 induced by HgCl_2_ occurs through a caspase-3 independent pathway. These results are confirmed in comprehensive transcriptome profiling of H1299 cells treated with HgCl_2_, and may useful strategy for the diagnosis and treatment of diseases caused by environmental toxicants.

## 4. Materials and Methods

### 4.1. Chemicals and Reagents

Mercury chloride (HgCl_2_; 316512) was obtained from Sigma–Aldrich (Oakville, Ontario, Canada). A 1M stock solution of HgCl_2_ was prepared and diluted in 100% ethanol (EtOH). We used 1, 5, 10, 50, and 100 μM of HgCl_2_ in RPMI (Roswell Park Memorial Institute) containing 10% fetal bovine serum (FBS). MTS assay kit (G3581, Madison, WI, USA) was purchased from Promega, and the LIVE/DEAD kit (L3224, Waltham, MA, USA) was purchased from Invitrogen. Apoptosis Detection kit (556547, San Diego, CA, USA) was obtained from BD Pharmingen^TM^. The Ki-67 antibody (ab15580, Cambridge, MA, USA), cyclins B1 (ab32053, Cambridge, MA, USA), cyclin D1 (Cat. No. 2978S, Beverly, MA, USA) were obtained from Abcam, Santa Cruz Biotechnology, and Cell Signaling, respectively. The secondary goat anti-rabbit antibody was purchased from Jackson ImmunoResearch (111-095-144, West Grove, PA, USA).

### 4.2. Cell Culture

All reagents for cell culture were obtained from Welgene (Seoul, Korea). The human non-small cell lung cancer cell line, H1299, was obtained from the Korean Cell Line Bank (KCLB) (Seoul, Korea). H1299 cells were grown in RPMI supplemented with 10% FBS and 1% penicillin or streptomycin at 37 °C in an incubator in an atmosphere of 5% CO_2_. At first, 0.1 mg/ml of poly-D lysine was placed on coverslips for 6 h at 23 °C, then H1299 cells were seeded in 100 mm culture dishes at a density of 2 × 10^6^ cells/dish. After 24 h, various concentrations of HgCl_2_ were treated in H1299 cells for 24 and 48 h. The cells on coverslips were used for the LIVE/DEAD cell assay, and the cells in the 10-cm culture dishes were processed for further experimental analyses. For each assay, 0.6% EtOH (*v/v*) was used as the negative control.

### 4.3. MTS Assay

H1299 cells were seeded at a density of 2 × 10^4^ cells/well in 48-well plates and were subsequently treated with different concentrations of HgCl_2_ in 100 μL RPMI supplemented with 10% FBS for 24 or 48 h. The H1299 cells were treated with 10% MTS (3-(4,5-dimethylthiazol-2-yl)-5-(3-carboxymethoxyphenyl)-2-(4-sulfophenyl)-2H-tetrazolium) and incubated for 1 h at 37 °C. Cell viability was assessed by measuring the absorbance at 490 nm using a Multiskan GO microplate reader (Waltham, MA, USA).

### 4.4. Morphological Change and Live or Dead Cell Assay

H1299 cells were seeded on coated coverslips and treated with different concentrations of HgCl_2_ for 24 and 48 h. The morphological changes were observed using an inverted microscope. For live or dead cell assay, H1299 cells were double-stained with 3 μM Ethidium Homodimer-1 (EthD-1) and 0.3 μM calcein-AM mixture (Life Technologies, Waltham, MA, USA) for 30 min in the dark condition. The morphology of the unwashed cells was observed using Nikon Eclipse TE300 Inverted Fluorescence Microscope (Nikon Corp., Tokyo, Japan).

### 4.5. Cell Cycle Analysis

H1299 cells were collected and were fixed in 70% ethanol with rotator for 1 h at 4 °C. Cell pellets were resuspended in 1× PBS (phosphate-buffered saline) containing 0.25 μg/μL RNase A and incubated for 1 h at 37 °C. The cells were treated with 10 μg/mL PI and incubated for 15 min in the dark condition at 23 °C. Finally, the cells were added 300 μL 1X PBS and then were analyzed using a BD Accuri™ C6 Plus flow cytometer (BD FACS, Franklin Lakes, NJ, USA). A minimum of 10,000 cells were considered per sample and the results are represented as histograms for analyzing cell distribution in the different phases of the cell cycle. The cell cycle profile was analyzed using BD Accuri™ C6 Plus software (BD FACS, Franklin Lakes, NJ, USA).

### 4.6. Annexin V-FITC and Propidium Iodide (PI) Apoptosis Assay

An FITC Annexin-V Apoptosis Detection Kit I (BD pharmingenTM, La Jolla, CA) was used for determining cellular apoptotic cell death according to the manufacturer’s instructions. Briefly, cells were collected and resuspended in 1X Annexin-V binding buffer [140 mM NaCl, 2.5 mM CaCl_2_, and 10 mM HEPES/NaOH (pH 7.4)]. Next, the cells were double-stained with 5 μL PI and 5 μL Annexin-V Alexa Fluor 488 and incubated for 15 min in the dark condition and were washed with 1X Annexin-V binding buffer. At least 10,000 cells were considered per sample and apoptosis was quantified by the BD Accuri™ C6 Plus flow cytometer (BD FACS, San Jose, CA, USA).

### 4.7. Immunostaining for Fluorescence-Activated Cell Sorting (FACS) Analysis

Cells treated with mercury chloride were fixed in 1% paraformaldehyde for 5 h on a rotator at 4 °C and centrifuged at 3000 rpm for 3 min. Each sample was resuspended with solution A (75 mM sodium acetate, 0.1% saponin, 0.1% BSA, and 25 mM HEPES, pH 7.2) and was incubated in the diluted indicated antibodies against IgG (1:5000), cyclin B1 (1: 200), cyclin D1 (1:200), Ki-67 (1:400), and cleaved capase-3 (1:200) for 1 h at 23 °C. After washing with 1X PBS, cells were incubated with FITC-labeled goat anti-rabbit secondary antibody (1:200) for 30 min in the dark at 23 °C. The cells were then washed three times in washing buffer and suspended in 200 μL of PBS. Forward scatter characteristics (FSC-A) and side scatter characteristics (SSC-A) plots were used to set the gates to distinguish between viable and non-viable cells. Each plot represents 10,000 viable cells (non-viable cells were excluded). For each experiment, FITC-A and SSC-A plots were used and IgG controls were used to set the gating parameters. The FITC signal was detected at 533 nm in the FL-1 channel. All data analyses were carried out using BD Accuri™ C6 Plus flow cytometer (BD FACS, CA, USA). A minimum of 10,000 cells were considered per sample and the results are represented as histograms. The percentage of the cell population was analyzed using BD Accuri™ C6 Plus software.

### 4.8. Total RNA Isolation

After treatment of HgCl_2_ for 48 h, cells were harvest at 3000 rpm for 3 min. The solvent 0.6% EtOH treated cells were used as control. Total RNA was extracted using TRIzol. The amount of total RNA was determined using Thermo scientific Multiskan GO microplate reader with μDrop™ Plate (Waltham, MA). The quality of total RNA was evaluated using the Agilent 2100 Bioanalyzer using the RNA 6000 Nano Chip (Agilent Technologies, Santa Clara, CA, USA). The RNA used in RNA-seq was considered to be high based on a RIN value of 7.

### 4.9. RNA-Seq Library Preparation and Sequencing

The construction of RNA-sequencing library was performed using Truseq RNA Sample preparation kit v2 (Illumina, Inc., San Diego, CA, USA, RS-122-2002). In briefly, 1 μg of total RNA (100 ng of mRNA) from each sample was incubated with poly-T oligo-attached magnetic beads to isolate poly-A tailed mRNA followed by mRNA fragmentation. The cleaved RNA fragments were constructed a double-stranded cDNA. Then, the double-stranded library was purified by using AMPure XP beads to remove all reaction components. The end repair, A base addition, adapter ligation, and PCR amplification steps were experimented according to the manufacturer’s instructions. Libraries quality and quantity were evaluated by an Agilent 2100 Bioanalyzer using the High Sensitivity DNA Chip (Agilent Technologies, Santa Clara, CA, USA). Then, the cDNA libraries were used for the paired-end 75 sequencing using an Illumina HiSeq 2500 (Illumina, Inc., San Diego, CA, USA).

### 4.10. Bioinformatical Analysis

Raw sequence reads were trimmed for adaptor sequence and masked for low-quality sequences using Trimgalore (ver. 0.6.5). Additionally, mRNA-Seq reads were aligned using STAR (ver. 2.5.4b) by reads aligned to Ensemble v98 Homo Sapiens transcriptome annotation (GRCh38.98). Transcript quantification of mRNA-seq reads was performed with Rsubread (ver. 2.0.1). The FPKM values were calculated using ‘fpkm’ function from DESeq2 (ver. 1.26.0) that processed on the robust median ratio method and transcript reads were normalized by ‘voom’ function from Limma (ver. 3.42.2). To analyze a transcript as differentially expressed, EdgeR (ver. 3.28.1) calculates the results based on the normalized counts from entire sequence alignments. Significantly differentially expressed transcripts having greater than log2 fold change >1 and adjusted *p*-value < 0.05 cases in all experimental comparison were selected and used for further analysis. Gene annotation was added by the online database using Ensemble biomaRt (ver. 2.40.4), and visualization was performed by using R base code and ggplot2 (ver. 3.3.2)

### 4.11. GO Searches

GOrilla (http://cbl118 gorilla.cs.technion.ac.il/; finally accessed at 1 Decemebr 2020) allowed significant in respective gene sets to be clustered according to their functional roles [37]. Enriched GO terms in ranked lists of genes by GOrilla were visualized using REVIGO bioinformatics resource [38]. All was selected for ontogeny choices and the *p*-value < 0.01.

### 4.12. Statistical Analyses

Data are presented as the mean ± standard error of the mean (S.E.M).; experiments were performed in triplicate. Data were analyzed using two-way ANOVA followed by Tukey’s multiple comparison test using GraphPad Prism5 software (California, CA, USA). Differences between the groups were considered to be significant at *p* < 0.05.

## Figures and Tables

**Figure 1 ijms-22-02006-f001:**
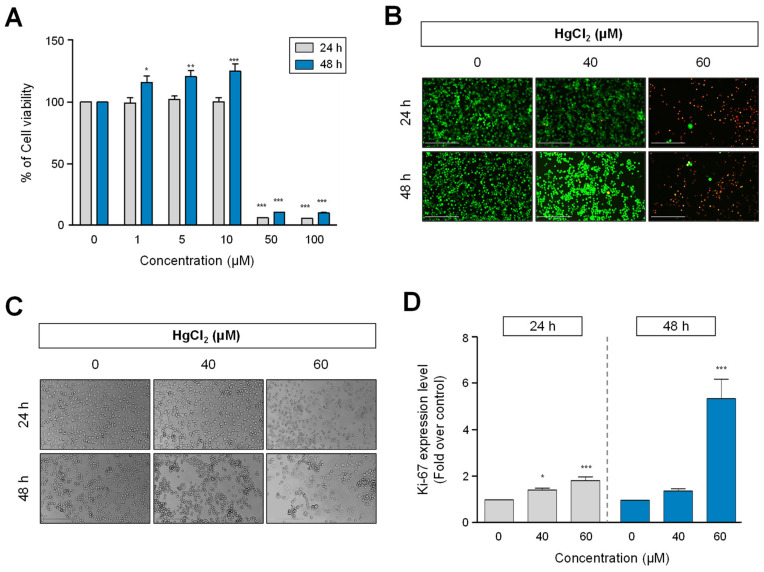
Cytotoxic effect of HgCl_2_ on human non-small cell lung carcinoma cells. (**A**) Cell viability was assessed by MTS (Tetrazolium inner salt) assay described in materials and methods. Cells were treated with different concentrations (0–100 μM) of HgCl_2_ for the indicated incubation time. The data expressed as a percentage of the values found in the solvent treated group (0.6% EtOH) used as control. * *p* < 0.05, ** *p* < 0.01, *** *p* < 0.001 compared with cells treated with 0.6% EtOH. **(B)** Cells treated with 40, 60 μM of HgCl_2_ treated for 24 or 48 h in H1299 cells. Cells were stained with calcein-AM (live cells, green) and ethidium homodimer (dead cells, red), according to the live or dead assay described in materials and methods. Scale bars= 200 μm. (**C**) Morphological changes of H1299 cells were observed by phase-contrast microscopy after HgCl_2_ treatment. Scale bars= 200 μm. (**D**) Cellular behavior by HgCl_2_ treatment assessed using FACS analysis. Cells were stained antibodies against each marker: Ki-67 for cell proliferation. Quantitative analysis of Ki-67 level induced by HgCl_2_ treatment was presented as percentage of Ki-67 positive cells. Values are presented as mean ± SEM, *n* = 6. * *p* < 0.05, *** *p* < 0.001 compared to cells treated with 0.6% EtOH.

**Figure 2 ijms-22-02006-f002:**
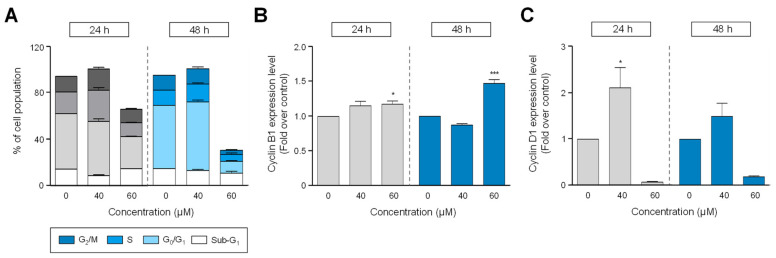
Alters in cell cycle progression of human non-small cell lung carcinoma cells following HgCl_2_ exposure. Cells were treated with 40, 60 μM of HgCl_2_ for 24 or 48 h. (**A**) After HgCl_2_ treatment, cells were fixed with 70% EtOH for 1 h. Cell cycle distribution was analyzed by FACS based on propidium iodide (PI) staining as described in materials and methods. Values are the percentage of each population in the Sub-G_1_, G_0_/G_1_, S, and G_2_/M-phase. To detect cell cycle-related markers, cells were fixed with 1% PFA for 6 hours. Then, cells were stained with the specific antibody against cyclin B1 (**B)** or cyclin D1 (**C**) for 30 min followed by flow analysis. Data are presented as the percentage of cyclin B1- (**B**) or cyclin D1 (**C**) positive cells and shown as mean ± SEM, *n* = 6. * *p* < 0.05, *** *p* < 0.001 compared with cells treated with 0.6% EtOH.

**Figure 3 ijms-22-02006-f003:**
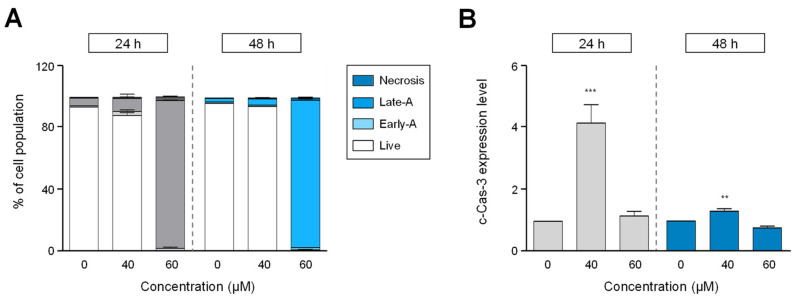
Exposure to HgCl_2_ induced apoptotic cell death in H1299 cells. The cells were incubated with indicated concentrations (40–60 μM) of mercury chloride for 24 or 48 h. (**A**) The cells were then stained with annexin V-FITC (Fluorescein isothiocyanate) and propidium iodide (PI) for apoptosis. The proportion of cells in apoptosis were measured by FACS. The bar graph is represented as the percentage of live cells in early and late apoptosis and necrosis. (**B**) An active form of caspase-3 was analyzed by the measurement of cleaved caspase-3. Cells were analyzed by flow cytometry in H1299 cells. Data are expressed as the percentage of cleaved caspase-3 positive cells and shown as mean ± SEM, n = 6. ** *p* < 0.01, *** *p* < 0.001 compared with cells treated with 0.6% EtOH.

**Figure 4 ijms-22-02006-f004:**
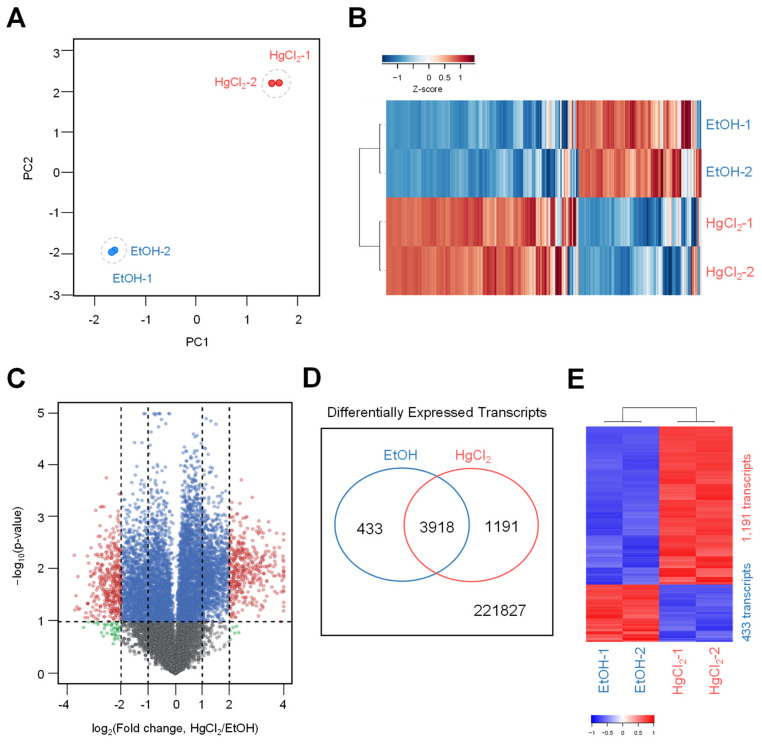
Comprehensive transcriptome dynamics by the HgCl_2_ treatment in H1299 cells. (**A**) Principal-component analysis (PCA) of directional RNA-seq data. Small circles indicate individual samples (Blue; EtOH, Red; HgCl_2_), and larger show each group. (**B**) Transcriptional pattern analysis in control and HgCl_2_-treated groups by employing heat map and hierarchical clustering. (**C**) Volcano plot indicates differentially expressed transcripts in HgCl_2_ treated group compared to the EtOH-treated group. (**D**) Differentially expressed transcripts were verified by the DEseq2 in the HgCl_2_-exposed group from total annotated transcripts. (**E**) Heatmap describing fold changes for all transcripts indicating statistically significant differences between the HgCl_2_-treated group and the control group.

**Figure 5 ijms-22-02006-f005:**
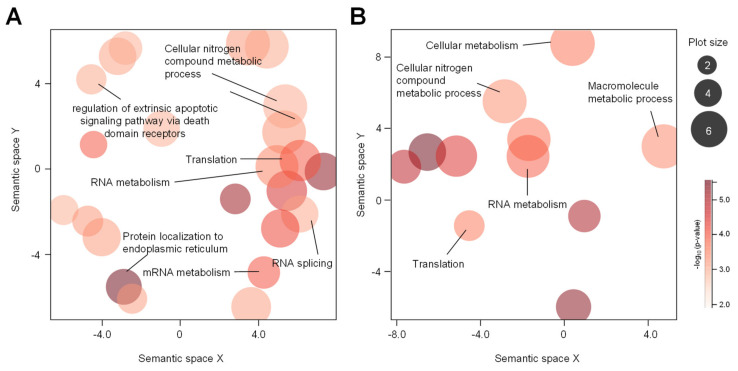
The categorization of gene ontology with the differentially expressed genes in the HgCl_2_-treated group. Gene Ontology terms from GOrilla were further visualized by REVIGO. Gene Ontology enrichment analysis of biological processes for up- (**A**) and down-regulated genes (**B**) between HgCl_2_- vs. 0.6% EtOH-treated group. Circle size reveals proportionally the frequency of the GO term, whereas color indicates the log10 (*p*-value) for the enrichment derived from the GOrilla analysis.

## Data Availability

The data presented in this study are available in the article or Appendix A. Also, raw sequencing fileof RNA-seq is avlailable at GSE163937.

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
