# Peer review of "Transcriptome Analysis Reveals HgCl2 Induces Apoptotic Cell Death in Human Lung Carcinoma H1299 Cells through Caspase-3-Independent Pathway"

_ijms, 2021, doi:10.3390/ijms22042006_

Round 1
Reviewer 1 Report
The paper is well written, really interesting and should be published but first needs to be clarified in some points.
1) HgCl2 role through ribosome impairment has already been shown (Chen M, von Mikecz A. Specific inhibition of rRNA transcription and dynamic relocation of fibrillarin induced by mercury. Exp Cell Res. 2000 Aug 25;259(1):225-38. doi: 10.1006/excr.2000.4923. PMID: 10942594 ; Vieira, J.C.S., de Oliveira, G., Braga, C.P. et al. Parvalbumin and Ubiquitin as Potential Biomarkers of Mercury Contamination of Amazonian Brazilian Fish.Biol Trace Elem Res
197, 667–675 (2020). https://doi.org/10.1007/s12011-020-02026-w
; Min Chen, Anna von Mikecz, Specific Inhibition of rRNA Transcription and Dynamic Relocation of Fibrillarin Induced by Mercury, Experimental Cell Research, Volume 259, Issue 1, 2000, Pages 225-238, ISSN 0014-4827,
https://doi.org/10.1006/excr.2000.4923 .) so I suggest to explain, both in the introduction and discussion, the importance of the present paper. I would also suggest to the authors that, while before the studies focused on few ribosomal genes, your research show a more general vision of the ribosome impairment. I still suggest the author to stress this point of view.
2)How the authors explain that Caspase 3 is overexpressed at lower
concentration?
Reviewer 2 Report
General: The manuscript aimed at discussing the toxic effect of inorganic mercury on lung cells. They used NSCLC cells as model and found the changes of cell viability, cell cycle and apoptosis in the treated cells. The authors also verified the transcriptome in this NSCLC cells to reveal the possible mechanism and the whole gene expression profiles of NSCLC cells by treating this inorganic mercury. There are several queries about this work as followed:
- There are many lung normal cell lines available for toxicological study, better than using cancer cells as model. Why did the authors use NSCLC cells as the model cells to test the toxicological effect of inorganic mercury, since this cell line is a lung cancer cell line?
- The authors used immune-fluorescent staining and flowcytometry to analyze the expression of Ki-67, caspase-3, cyclin D1 and B1. The parameters are the fluorescent strength of each protein and side scatter channel (SSC). The authors should explain why they use the granularity of the cells and what is the purpose of this parameter in the experiment?
- Ki-67 scoring was used the percentage of positive cells within total counted cells, rather than using the folds of control.
- Cell cycle distribution could be analyzed by automatic software such as ModFit, rather than using handmade gating.
- The threshold of positive/negative cells should be defined by using unstained cells and all positive cells. The authors should explain how their threshold was made.
Round 2
Reviewer 1 Report
The authors made all the suggested changes.
Author Response
We appreciate the reviewer's decision on our manuscript. As the reviewer mentioned, this study will give information on heavy metal toxicity.
Reviewer 2 Report
The authors responded to the question of major revision very quickly, but more questions were raised as follows:
- The authors agree that the main purpose of this paper is to explore the toxicity of mercuric chloride in lung cells, and also agree that the use of normal lung cells is correct. They are now trying to find easy-to-culture normal lung cells such as MRC-5 for experiments. Because of a major revision, the authors should complete the results of the toxic reaction of MRC-5 to mercuric chloride before proceeding with this revise work, rather than just giving an ongoing statement. It is strongly recommended that authors should complete the test results of MRC-5 cells and add them to the revised version.
- The authors emphasizes that the cells in flow cytometry immunoassay are all parts of living cells. However, there is no any definition in their methods and results how the living cells are judged and selected in flow cytometry. The cells used by the authors are the cultured homogeneous cells. Generally, the size (FSC) and intracellular granularity (SSC) of living cells will be in the defined narrow range. As shown in figure 1D, 0 and 40μM groups cells are concentrated in places where the SSC is less than 250 units. However, 60μM is obviously increased to more than 250 units. Figure 1 A and B have shown that the cell survival status at 60μM is very poor, how can the authors distinguish live cells from dead cells in the results of flow cytometry? It may be difficult to obtain appropriate results by immunostaining from such cells to determine their protein expression.
- The author's skill on cell cycle analysis is also very questionable. It may be technically unsatisfactory. The result of Figure 2A shows that the untreated cells have a different PI fluorescence intensity range defined in the G1 phase of 24 hours and 48 hours. The median value of 24 hours is approximate at 106, but at 48 hours, 106 is the upper limit of G1 cells. The untreated cells contained more than 10% sub- G1 cells in the sub-G1 region defined by the author. Does this mean that there is a problem with the author’s cell culture technology or the pre-processing of the cells, causing the cell staining intensity distribution of these cells to be so unstable? The author added the automatic detection result of cell cycle distribution with ModFit software. Comparing the results in the manuscript, the estimated distributions of G1 phase of untreated cells found in MotFit software at 24 and 48 hours are 61.3% and 75.1%, while the author's original is 49.3% and 59.5%. Moreover, the FL-2A (PI) intensity value seen from the ModFit graph, the G1 intensity position captured from each treatment group is not consistent, which means that the author's cell cycle analysis technology is not stable, and the reliability of the results is questionable.
Round 3
Reviewer 2 Report
The authors have done their efforts on explaining their results and their hard work in this manuscript. There are a few modifications and suggestion as followed:
- Figure 2A should be deleted because that is not necessary, indeed, to demonstrate the calculation results. If the authors hardly present consistent cell cycle histograms, data not shown is acceptable.
- There are two Figure 2C in Figure 2, please correct.
- All of the immunofluorescent staining by flowcytometry could be presented using single parameter histogram or deleted them. Because the living cell population is not judged by SSC (intracellular complexity). The authors have pre-selected the living cells by gating the homogenous cell population in FSC/SSC, the immune strength of each protein can be estimated on single parameter histogram by separated peak or peak shift. Also, the original flowcytometry figures are not the necessary results that must be shown in the manuscript. So I suggest the authors could consider to delete them.
